# First detection of Crimean Congo Hemorrhagic Fever antibodies in cattle and wildlife of southern continental France: Investigation of explanatory factors

Célia Bernard[1,2,3]*, Andrea Apolloni[1,2], Vladimir Grosbois[1,2], Armelle Peyraud[1,2], Phonsiri Saengram[1,2], Ferran Jori[1,2], Eva Faure[4], Nicolas Keck[5], Raphaëlle Pin[6], Olivier Ferraris[7], Loic Comtet[8], Benoit Combes[3], Matthieu Bastien[3], Valentin Chauvin[1,2], Laure Guerrini[1,2], Philippe Holzmuller[1,2], Laurence Vial[1,2]

**1** CIRAD, UMR ASTRE, Montpellier, France, **2** ASTRE, Univ Montpellier, CIRAD, INRAE, Montpellier, France, **3** French Establishment for Fighting Zoonoses (ELIZ), Malzéville, France, **4** National Hunting Federation, Montpellier, France, **5** Department Veterinary Laboratory of Hérault, Montpellier, France, **6** Department Veterinary Laboratory of Alpes-Maritimes, Nice, France, **7** Institut de Recherche Biomédicale des Armées, Brétigny-sur-Orge, France, **8** Innovative Diagnostics, Grabels, France

* celia.bernard@cirad.fr

## Abstract

Crimean-Congo Hemorrhagic Fever (CCHF) is a tick-borne zoonosis of major public health concern, not only because of its potential for severe outcomes in humans, but also due to its endemic presence in many regions and its expanding geographic distribution. We report on the first serological survey conducted in mainland France to detect antibodies against the Crimean-Congo Hemorrhagic Fever Virus (CCHFV) in domestic and wild fauna, and provides critical insights into the virus's circulation. We analyzed 8,609 cattle sera and 2,182 wildlife sera collected across the French Mediterranean region from 2008 to 2022, using enzyme-linked immunosorbent assays (ELISA) and pseudo-plaque reduction neutralization tests (PPRNT) for antibody detection and confirmation. Seropositivity was detected in both cattle (2.04%) and wildlife (2.25%), with higher rates observed in specific regions including the Pyrénées-Orientales and Hautes-Pyrénées. These findings reveal spatial clusters of CCHFV circulation and suggest the existence of enzootic transmission cycles involving local tick vectors and animal hosts. Our multivariate analysis identified key factors that influence seropositivity, including animal age, habitat characteristics, and potential wildlife interactions. The presence of natural open habitats and coniferous forests was significantly associated with higher seropositivity in cattle, while sex and geographical variability played a role in wildlife seroprevalence. These findings highlight the importance of environmental and anthropogenic factors in shaping the dynamics of CCHFV transmission. This work demonstrates that CCHFV is actively circulating in parts of mainland France, emphasizing the need for enhanced surveillance and integrated approaches to monitor zoonotic pathogens. It also raises questions about

**Data availability statement:** The files are available on this repository: https://gitlab.cirad.fr/astre/data_sero_cbernard2025.

**Funding:** The authors acknowledge the funders who made this work possible: French Ministry of Agriculture—General Directorate for Food (DGAI, grant agreement: SPA17 number 0079-E), European Funds for Regional Development (FEDER, Grand-Est), French Establishment for Fighting Zoonoses (ELIZ) and the Association Nationale Recherche Technologie (ANRT, grant agreement number: 2019-1145). The funders had no role in study design, data collection and analysis, decision to publish, or preparation of the manuscript.

**Competing interests:** The authors have declared that no competing interests exist.

the role of additional tick vectors, such as *Hyalomma lusitanicum*, in the transmission cycle. These results advance our understanding of CCHF epidemiology and offer valuable guidance for public health strategies to mitigate the risks associated with this emerging disease.

## Introduction

Crimean-Congo Hemorrhagic Fever (CCHF) is a tick-borne zoonosis caused by infection with a RNA virus of the *Orthonairovirus* genus and *Nairoviridae* family [1]. CCHF is one of the most widespread tick-borne diseases and has the widest worldwide distribution [2] CCHF occurs in Africa, the Middle East, the Balkans, and Central Asia [3,4]. Its recent detection in new areas including India and Pakistan, as well as western Europe in Greece and Spain over the last two decades, is a public health concern [3,5,6]. Humans are infected through tick bites, or when they are exposed to the blood or infected tissues of viremic animals, or the blood of infected people [2]. Not all infected people develop symptoms, these are a first phase of fever, tremors, myalgia, headache, nausea and vomiting, abdominal pain and arthralgia, and sometimes complications with internal and external bleeding as well as ecchymosis; mortality of infected people ranges from 9% to 50%, depending on the time of treatment and the virulence of the viral strain [7,8]. In contrast, infection with Crimean-Congo hemorrhagic fever virus (CCHFV) does not cause any clinical signs in either wild or domestic animals, although most of infected species are capable of becoming viremic and of developing a humoral immune response [9]. Although viremia in animals is short (5–10 days), antibodies against CCHFV are maintained for several years particularly in large ungulates, making serology an excellent surveillance tool to measure animal exposure to CCHFV and to detect early virus circulation in a naïve area [9,10]. In addition, animals can play essential roles in CCHFV transmission, by either increasing tick vector populations, the transient replication of CCHFV to reinfect new tick vectors, or the spread of infected ticks [11,12]. Animals can only be infected by being bitten by infected ticks, and this method of circulation of the virus between vertebrate animals and ticks is the natural enzootic transmission cycle of CCHFV [10]. The CCHFV genome has been detected in many tick species worldwide but only ticks of the genus *Hyalomma* (with *H. marginatum* and *H. lusitanicum* as likely vectors in southern Europe) are considered to be the most competent vectors for CCHFV [13,14]. Given ticks' ability to remain infected during their development cycle, and their competence not only for horizontal but also transovarial transmission, as well as cofeeding, ticks are also considered to be natural reservoirs of CCHFV [15]. Consequently, the geographical distribution of CCHFV could expand through the introduction and establishment of infected ticks in new suitable territories, by either migratory birds or by human-driven long-distance movement of livestock. Since 2010, CCHF has been detected in Western Europe, the virus genome was detected in *Hyalomma spp.* ticks in Spain [16,17] and a few human cases linked to tick bites have been reported since 2016 [18]. Antibodies against CCHFV have been detected

in Spanish cattle and wildlife in Spain [19,20] but also in bovine sera from southern Italy [21]. In France, a serological survey was conducted on domestic ruminants in Corsica, a French island located off the southeast coast of the mainland, between 2014 and 2016, and detected about 9% of positive animals (cattle, sheep and goat) among the 4,000 tested (22). Given the recent establishment of a known tick vector (*H. marginatum*) in southern coastal areas of the mainland [22,23], the epidemiological situation of CCHF in this area is also questioned. Recently, CCHFV was detected for the first time in *H. marginatum* ticks on the mainland in Pyrénées-Orientales, as well as on the island of Corsica, confirming the local transmission of CCHFV in France [24,25]. Other tick species such as *Rhipicephalus bursa* and *Dermacentor marginatus* have occasionally been found carrying CCHFV in southwestern Europe, but their role in transmission remains uncertain. In France, despite the presence of multiple tick genera, only *H. marginatum* has been found positive to date. Because the prevalence of tick infection is assumed to be very low in French tick vectors due to a dilution effect [26], monitoring CCHFV in ticks remains laborious and compared to serology, requires large-scale sampling to identify virus circulation areas. In addition, knowing where tick vectors are located is necessary but not sufficient to identify areas of virus circulation, and other factors need to be investigated. No studies have been carried out in Corsica to identify the factors that influence seroprevalence, but such research has been carried out in other countries, and identified a number of factors that impact either the exposure of animals to infected tick vectors or the level of CCHFV transmission within the enzootic cycle [27,28].

In the present study, we focused on the French mainland where no serological survey has ever been carried out. Sera from cattle and wildlife from different areas of the French Mediterranean rim were analyzed to detect antibodies against CCHFV and to estimate individual seroprevalence in different animal populations as indicators of current levels of CCHFV circulation. Using a correlative approach, we also aimed to identify individual, environmental and anthropogenic determinants of CCHFV transmission.

## Materials and methods

### Study area/period and sample selection

The study area focused on south-eastern France from the Spanish to the Italian border, in areas where the presence of *H. marginatum* has been confirmed, or nearby areas predicted to be suitable for its establishment in the more or less long term due to climate change [22,23] (Fig 1). Some neighboring locations where *H. marginatum* is *a priori* absent were also selected, as the process of establishment of this exotic tick species is ongoing [23]. Sampling was carried out at the level of the department (corresponding to the NUTS-3 administrative division in the European nomenclature) and at the level of the municipalities (lowest division in a department). Most of these departments host a large number of cattle, particularly suckling calves raised for the meat market. A number of local breeds also exist, for example bulls and cows raised in the Camargue, a humid area in the Rhône delta for Camargue bullfights and associated recreational activities. Wild boar is common throughout the study area, while cervids including roe and red deer are numerous in the dense forests of the hinterland.

Sera from cattle were selected among those collected during the national preventive program. Indeed, as part of French national disease control measures, the blood of domestic livestock (large and small ruminants) is sampled every winter for serological testing for several pathogens. Blood samples are collected by official veterinarians in each French department, and sera are analyzed by official veterinary departmental laboratories. For our study, we thus needed the laboratories' agreement to take part and to give us the samples after use, while selection of samples we received depended on the laboratories' own sample management and storage policy. Given this primary constraint, subsampling of sera was conducted in order to obtain broad spatial coverage in the departments concerned and to be able to detect positive farms and estimate animal seroprevalence, at least at the municipal level. As *H. marginatum* is distributed in spatial clusters and appears to be continuing to spread [23], we predicted small local foci of CCHFV circulation – if any exist – with no assumption concerning the level of spatial heterogeneity. For this reason, when possible, all municipalities with cattle

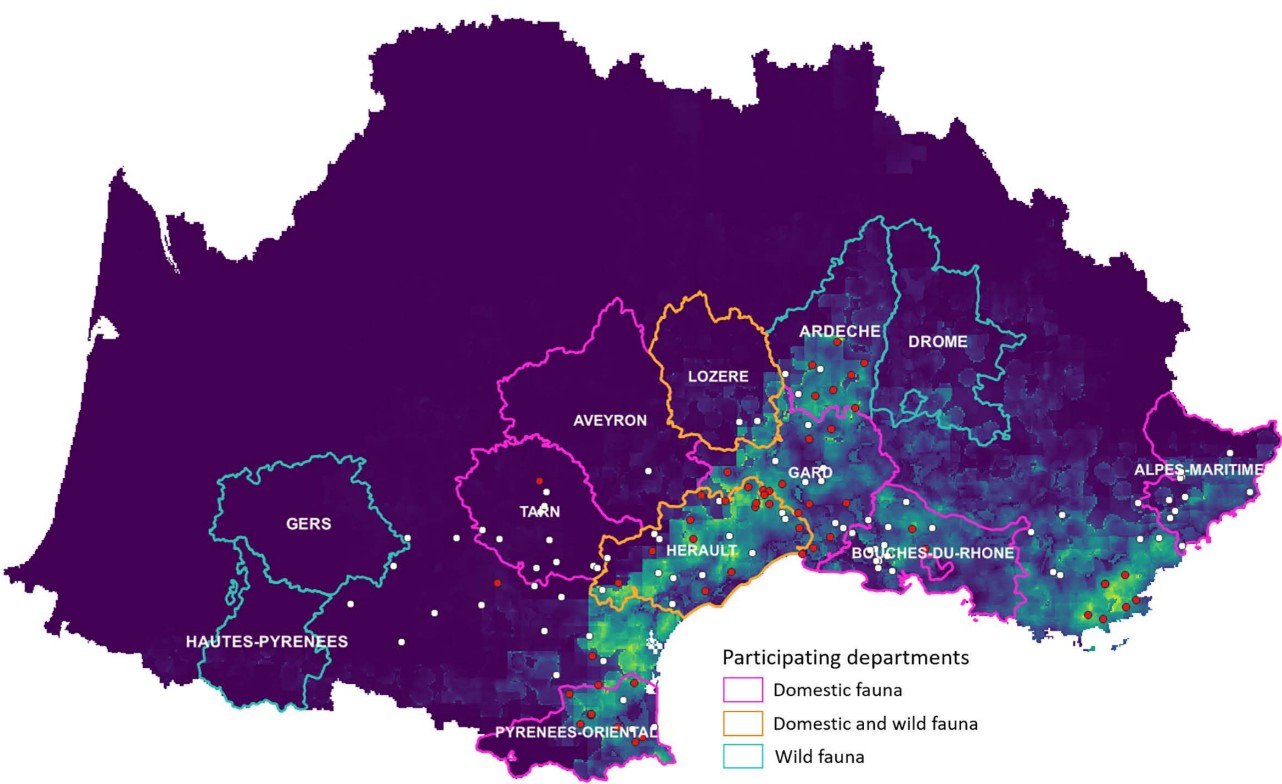

**Fig 1. Study area showing domestic and/or wild fauna in the administrative departments that agreed to take part in the study.** In the background, the map predicted probability of presence for *H. marginatum* in the South of France, using the most parsimonious model (dark blue: null probability to bright yellow: highest probability). Red dots represent the observed presences and white dots observed absences from testimonies and surveys. Adapted from [23].

farms in participating departments were included in our survey and all the farms located in these municipalities (or at least 6 farms per municipality) were tested. In the previous serological investigation in Corsica, at least one positive cow was identified on more than 25% of farms (Grech-Angelini, personal comm.), but we expected a much smaller proportion of seropositive farms on the mainland since *H. marginatum* had only recently been established. The number of sera was limited to 5–10 animals on each farm, which was considered enough to conclude on the epidemiological status of farms, as the estimated individual seroprevalence in positive farms in Corsica was higher than 30% (Grech-Angelini, personal comm.). Sera were selected from three successive prevention campaigns (2018–2019, 2019–2020, and 2020–2021) (Table 1) to cover as many farms as possible, depending on the samples available in the laboratories. A second prevention period (2021–2022) was added to complete our sera sampling campaign but only in Pyrénées-Orientales. Because of the long persistence of CCHFV antibodies in most vertebrate animals [9], and the fact that prevention campaigns differed from one department to another, in our analysis, we considered all the successive years as a single period in order to enable comparison. In the case of the few animals that were sampled twice (N = 262), we only took their last serological status into account, as no animal became seronegative from one year to the next and only three animals seroconverted.

Obtaining sera from wildlife was facilitated by the national hunter's federation in some departmental hunting associations that agreed to take part in the study (Fig 1). Collections of sera were available for several hunting seasons between 2008 and 2022, and included samples from wild boar (*Sus scrofa*), mouflon (*Ovis orientalis*), red deer (*Cervus elaphus*), roe deer (*Capreolus capreolus*), and fox (*Vulpes vulpes*) (Table 2).

**Table 1. Number of cattle, municipalities, and farms analyzed to detect CCHFV antibodies in each French administrative department that agreed to take part in the study, and number of samples of cattle sera found positive in CCHFV antibodies.**

| Department | No of sera tested | No of farms tested | No of positive results | Year of sampling |
|---|---|---|---|---|
| **Alpes-Maritimes** | 571 | 75 | 41 (7 farms) | 2019–2020 |
| **Aveyron** | 1246 | 126 | 2 (2 farms) | 2019–2020 2020–2021 |
| **Bouches-du-Rhône** | 1710 | 188 | 7 (3 farms) | 2018–2019 2019–2020 2020–2021 |
| **Gard** | 1013 | 60 | 17 (2 farms) | 2018–2019 2019–2020 |
| **Hérault** | 1946 | 191 | 28 (10 farms) | 2018–2019 2019–2020 |
| **Lozère** | 100 | 20 | 0 | 2020–2021 |
| **Pyrénées-Orientales** | 880 | 87 | 80 (19 farms) | 2019–2020 2021–2022 |
| **Tarn** | 1112 | 111 | 1 (1 farm) | 2019–2020 2020–2021 |
| **Other (Var and Vaucluse)** | 31 | 4 | 0 | 2019–2020 |

**Table 2. Number of samples of wildlife sera obtained between 2008 and 2020 for each animal species hunted that were analyzed to detect CCHFV antibodies in each French administrative department that collected sera from hunted animals and that agreed to take part in the study, plus the number of positive sera found in CCHFV antibodies.**

| Department | No of sera tested | No of munici-palities tested | No of positive results | Wild boar | Roe deer | Red deer | Mou-flon | Fox | Year |
|---|---|---|---|---|---|---|---|---|---|
| **ARDÈCHE** | 28 | 5/335 | 0 | 0/17 | 0/11 | – | – | – | 2009–2012 |
| **DRÔME** | 434 | 35/363 | 0 | – | 0/434 | – | – | – | 2016–2020 |
| **GERS** | 379 | 98/461 | 0 | 0/74 | 0/305 | – | – | – | 2010–2021 |
| **HAUTES-PYRENEES** | 621 | 148/469 | 45 | 9/179 | 16/197 | 20/244 | 0/1 | – | 2008–2022 |
| **HÉRAULT** | 622 | 49/342 | 3 | 3/566 | 0/23 | 0/8 | 0/21 | 0/4 | 2012–2022 |
| **LOZÈRE** | 98 | 12/152 | 1 | 0/33 | 0/7 | – | 1/58 | – | 2019–2022 |

## Serological analysis of samples

**Enzyme-linked immunosorbent assay (ELISA).** All sera selected from cattle and wildlife were tested for the presence of IgG and IgM antibodies against CCHFV, using a double-antigen ELISA kit (ID Screen CCHF Double Antigen Multispecies, Innovative Diagnostics (formerly IdVet, https://www.innovative-diagnostics.com) according to the manufacturer's instructions. For this kit, the 95% CI for sensitivity is 96.8%–99.8%, and the 95% CI for specificity is 99.8%–100% [29]. A freeze-dried serum provided by the manufacturer (MRI-CCHF) was used as an external control to ensure that analytical sensitivity remained constant between runs and to determine the test uncertainty. Briefly, after reconstitution, the serum was serially diluted to obtain the detection threshold. Aliquots (maximum 3 freezing/thawing cycles per aliquot) were prepared and analyzed on each test run. For each serum sample, the absorbance of the colorimetric immuno-enzymatic reaction of the sample was divided by the absorbance of the positive control to obtain a percentage as the optical density (OD) of the sample. Samples with OD < 30% were considered negative and those with OD ≥ 30% were considered positive.

**Pseudo-plaque reduction neutralization (PPRN).** To confirm the serological results obtained using the ELISA test, we sent our ELISA-positive sera as well as 10 ELISA-negative sera (5 from Corsica and 5 from the CCHFV-free Netherlands) to a Biosafety Level 4 laboratory (Laboratory Jean Mérieux, Lyon, France) to be analyzed by the national reference laboratory for CCHFV, (*Institut de Recherche Biomédicale des Armées,* Paris, France) designated by the World Organization for Animal Health. They used the Pseudo–Plaque Reduction Neutralization test (PPRNT) [6] to measure the neutralizing effect of sera antibodies against the IbAr10200 CCHFV strain (the same antigen as the one used in the ELISA test), at successive dilutions 1/20, 1/40, 1/80, 1/160. Results are given relative to the

dilution threshold (DT) until which the neutralizing effect of each sera sample was observed. We also included in our confirmation process 5 ELISA-positive Corsican cattle sera for which the absence of immune cross-reaction with CCHFV close-related viruses (namely, the Hazara virus from the same serogroup as CCHFV and the Dugbe virus that belongs to the close Nairobi sheep disease serogroup) was confirmed in a previous CCHF survey, although they clearly neutralized CCHFV culture [30].

**Statistical analysis of serological data.** Given the importance of protecting the farmers' personal data and the risk of stigmatizing some farms, it was not possible to present serological results found in cattle for each individual farm. In addition, taking into account the variability of the availability of samples of sera across localities, it would have been biased to present data for each municipality and so we chose to use Voronoi polygons that make it possible to mask administrative divisions and to delineate new regional boundaries in agreement with the serum sampling effort. This innovative approach facilitated estimation of seroprevalence distribution and was conducted using the Voronoi function in QGIS. This technique involved generating Voronoi polygons, which assign regions to sampled points based on proximity, effectively avoiding the influence of administrative boundaries. The QGIS tool provided the flexibility to create these polygons efficiently, ensuring a relatively unbiased visualization of the data distribution across the study area.

Prior to the statistical modeling, we constructed a Directed Acyclic Graph (DAG) to represent hypothesized causal relationships between individual, environmental, and anthropogenic variables and the serological status of animals. This approach helped identify potential confounding structures and guided the selection of variables to be included in the multi-variable models. The DAG is provided as S1 Fig.

In order to identify plausible factors linked to animal seropositivity against CCHFV in France (previously detailed in [26]), we modeled the individual status of animals against their own characteristics, as well as against environmental and anthropic factors. Individual information and rearing practices were extracted from the National Identification Database (French acronym BDNI) for cattle, and individual characteristics from hunting datasheets for wildlife, while environmental data were downloaded from data previously compiled by Bah et al. [23].

Concerning individual characteristics, for cattle, we included information on their sex (male or female), age (in years) at blood collection, and breed. For wildlife, we included information on their sex (male or female), age (estimated as young, subadult or adult) as well as the species. The year the animals were sacrificed was also mentioned, as the sera analyzed in this study were collected over a period of 14 years, and CCHFV transmission as well as host dynamics could have changed during the period.

Concerning rearing practices, for each farm representing the origin of cattle tested, we included information on the rearing of other domestic animals (i.e., sheep, goats, pigs, or a mix of these species) in addition to cattle on the farm, as an index of contacts between the cattle and other animal species that may have distinct abilities to replicate CCHFV and infect tick vectors. Cattle breeds were also grouped to create categories that reflect the different types of production, for which we assume distinct grazing practices and different exposure to CCHFV tick vectors (dairy cattle mostly in barns, rustic dairy cattle in barns and pastures, free-ranging suckling calves for meat production, specific cattle breeds bred for recreational purposes such as Camargue bullfights and associated activities). Concerning environmental data, the probability of the presence of *H. marginatum* in the South of France, derived from the predicted suitability of local habitats and climate for the establishment of this tick species [23], was the first information to be collected as an index of the animals' exposure to CCHFV tick vectors. However, we also considered an alternative scenario in which *H. marginatum* would not be the only – or would be the main – vector of CCHFV in France, which appears to be the case in Spain [31], and we consequently also collected raw climate and habitat data as an index of the presence of a tick vector. Indeed, each tick species has its own environmental needs to survive and complete its development cycle, which in turn, defines its spatio-temporal ecological niche [32]. In addition, climate and/or habitats may impact not only the dynamics of tick vectors but also other parameters of CCHFV transmission such as the distribution, the abundance or the specific diversity of vertebrate animal hosts, as well as the success of virus replication in ectothermic tick vectors [26]. For climate, we used

synthetic variables previously defined by principal component analysis, which processed meteorological data for the South of France for the period 2000–2018, including temperature, rainfall, potential evapotranspiration and relative humidity [23]: Dim 1 (positively correlated with lower temperatures all year round and higher summer rainfall); Dim 2 (negatively correlated with high relative humidity and winter rainfall, positively correlated with evapotranspiration); Dim 3 (positively correlated with rainfall in the fall). For habitats, we used a raster file (in pixels) representing 7 land cover classes (humid areas such as inland marshes and rice fields, open natural areas such as sclerophyllous vegetation and sparsely vegetated areas, open forests, broad-leaved forests, coniferous forests, urban or peri-urban areas, agricultural areas) generated by Bah et al. [23], through the combination of CORINE Land Cover (CORINE Land Cover 2018, Version 2020_20u1) and BD forêt (BD Forêt® version 2.0). To attribute climate and habitat, as well as the predicted probability of the presence of *H. marginatum*, rather than considering each farm on which cattle were serologically tested, we preferred considering the area in which they could graze rather than a local point represented by the farmstead. To this end, using the GPS coordinates of the farm's main building, we created a 4-km radius buffer zone around each GPS coordinate. Inside this buffer, we calculated the average probability of the presence of the *H. marginatum*, and mean values for the three climatic principal components. For the habitat, we calculated a proportion for each of the 7 classes in each buffer zone. As data for wildlife were located at the municipal level, similar values were calculated for climate, habitat and the probability of presence pf *H. marginatum* in each municipality in which wild animals were serologically tested. QGIS software (3.28.3-Firenze) ("Zonal statistics" and "Zonal Histogram") was used to process the spatial dataset.

To estimate the effect of these factors on the probability of an individual animal being seropositive, we built a Hierarchical Multivariable Logistic Regression Generalized Mixed Effect Model, one for cattle and one for wildlife, with a logit link function. The first model concerned cattle and had several nested levels (department, municipality within the department and farm within the municipality) and the breed (30 reported breeds) as random effects, and the other explanatory variables (sex, age, possible on-farm contacts with other animal species, pasturing practices, the probability of the presence of *H.* marginatum, climate, and habitat) as fixed effects. The second model concerned wildlife, with two levels of random effects (department, and municipality within the department), and the other explanatory variables (sex, age, year of hunting, the probability of presence *H. marginatum*, climate, and habitat) as fixed effects.

The models were processed in R (Rstudio, version 4.2.1). We used a dredge procedure (MuMin package) adding individual, anthropogenic and environmental fixed factors to select the most parsimonious model and only retain important explanatory variables. The optimal model was chosen using a combination of evaluation criteria: the Akaike information criterion (AIC) to evaluate the overall quality of the model, the percentage of data variance explained by the model, and we used the Somers' Dxy rank correlation test (Packages: car, MuMin, Lme4) to evaluate the predictive capability of the model, and the AUC (area under the ROC curve) to assess its explanatory capacity. Taking these different measures into consideration, the model that demonstrated the best balance between data fit, predictive ability, and explanatory power was selected as the top choice. We performed correlation tests on the variables retained in our final models to ensure there was no correlation between the variables.

### Ethic statement

Our study was strengthened by the voluntary participation of livestock farmers, who gave their written informed consent for the analysis of serum samples collected from their animals used to detect CCHFV antibodies. In most departments, farmers were individually informed about the purpose of the study, the confidentiality of the results, and the absence of any consequences for farms hosting seropositive animals. In the Tarn department, the *Groupement de Défense Sanitaire* (GDS), the local farmers' representative body, provided consent on behalf of the farmers for the analysis. To protect participant confidentiality, strict data protection protocols were followed during sample collection, storage, and analysis. Ethical considerations also covered the use and conservation of the samples, in accordance with the guidelines established in collaboration with CIRAD's legal department

## Results

### Identification of seropositive animals using ELISA

We obtained 8,609 samples of cattle sera for four disease prevention periods from 2018 to 2022 (Table 1), and 2,182 samples of wildlife sera from 2008 to 2022 (Table 2) to be tested using the ELISA test.

ELISA detected antibodies against CCHFV in 176 samples of cattle sera in most of the departments tested, with the exception of Lozère (Table 1 and Fig 2). Tarn had only one seropositive animal and Aveyron had only two, each isolated on a farm in a different municipality. The two seropositive cattle in Aveyron are animals that changed farms in the course of their lives: the first was born in Hérault in 2016 and arrived at the farm where it was sampled at the end of 2018. The second was born in Aveyron in 2011, and although it changed farms, it has remained in the north-west of the department throughout its life. The seropositive animal sampled in Tarn has never changed farm since it was born. Although individual seroprevalences were higher in Bouches-du-Rhône (0.4%), Hérault (1.44%) and Gard (1.68%), positive municipalities remained spatially dispersed and most farms contained only one seropositive animal, except four farms that contained several seropositive heads of cattle (intra-farm seroprevalence in farms with more than one positive animal 26.08% [4.08–66.67%]) (Fig 2). Individual seroprevalences were highest in Alpes-Maritimes (7.18%) and Pyrénées-Orientales

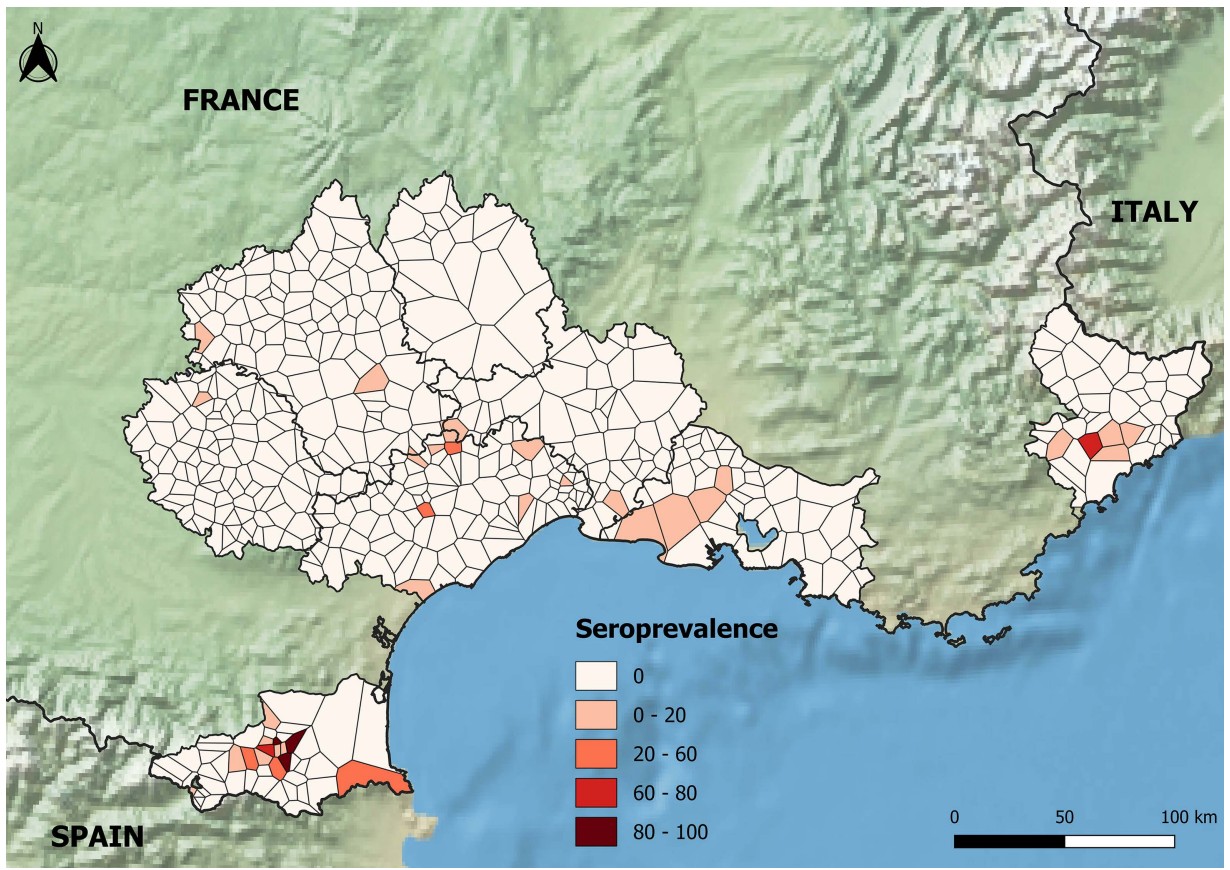

**Fig 2. Map of cattle individual seroprevalences per municipality (from pale pink to dark red the colors indicate increasing seroprevalence; sampled but negative municipalities are in white), in the administrative departments that agreed to take part in the study.** Voronoï polygons are used to represent the municipalities to guarantee the anonymity of farmers in sampled municipalities and to deal with the absence of sampling in some municipalities. Background layers derived from Natural Earth (CC BY 4.0).

(9.09%), where seropositive municipalities were clustered around apparent "hotspots" of transmission (intra-municipality seroprevalence 39.1% [11.11–100%]) and most of the positive farms contained many seropositive animals (intra-farm seroprevalence in clusters 47.93% [10–100%]) (Fig 2).

For wildlife, 49 positive samples were detected in wild boars (n = 14), red deer (n = 18), roe deer (n = 13) and mouflon (n = 1) (Fig 3). Among the 49 positive animals (including wild boar, roe deer and red deer) 45 were hunted in Hautes-Pyrénées (Table 2). Antibodies have been detected in Hautes-Pyrénées almost every year since 2008, except during the COVID years (2019–2021) when hunting pressure decreased dramatically.

Concerning the optical density (OD) values of samples of cattle sera tested by ELISA, positive and negative samples were grouped in two distinct populations (Fig 4A). Negative samples clustered around a consistent OD value (median = 3.68), while positive samples had two discernible profiles. Pyrénées-Orientales and Hérault both had homogeneous high OD values (median = 152.36). Conversely, OD values for the other positive departments were more widely distributed, with some samples approaching the ELISA threshold while others obtained higher OD values (median = 65.87). Among these samples, Alpes-Maritimes had a single prominent peak of OD values not far from values obtained in Pyrénées-Orientales and Hérault, while Gard had two distinct peaks. In the other departments, it was difficult to determinate profiles as positive samples were scarce but most of the OD values were low, close to the threshold. In wildlife, the OD values of all the negative samples were similarly low (median = 5.93) (Fig 4B). Like for cattle, the OD values of positive

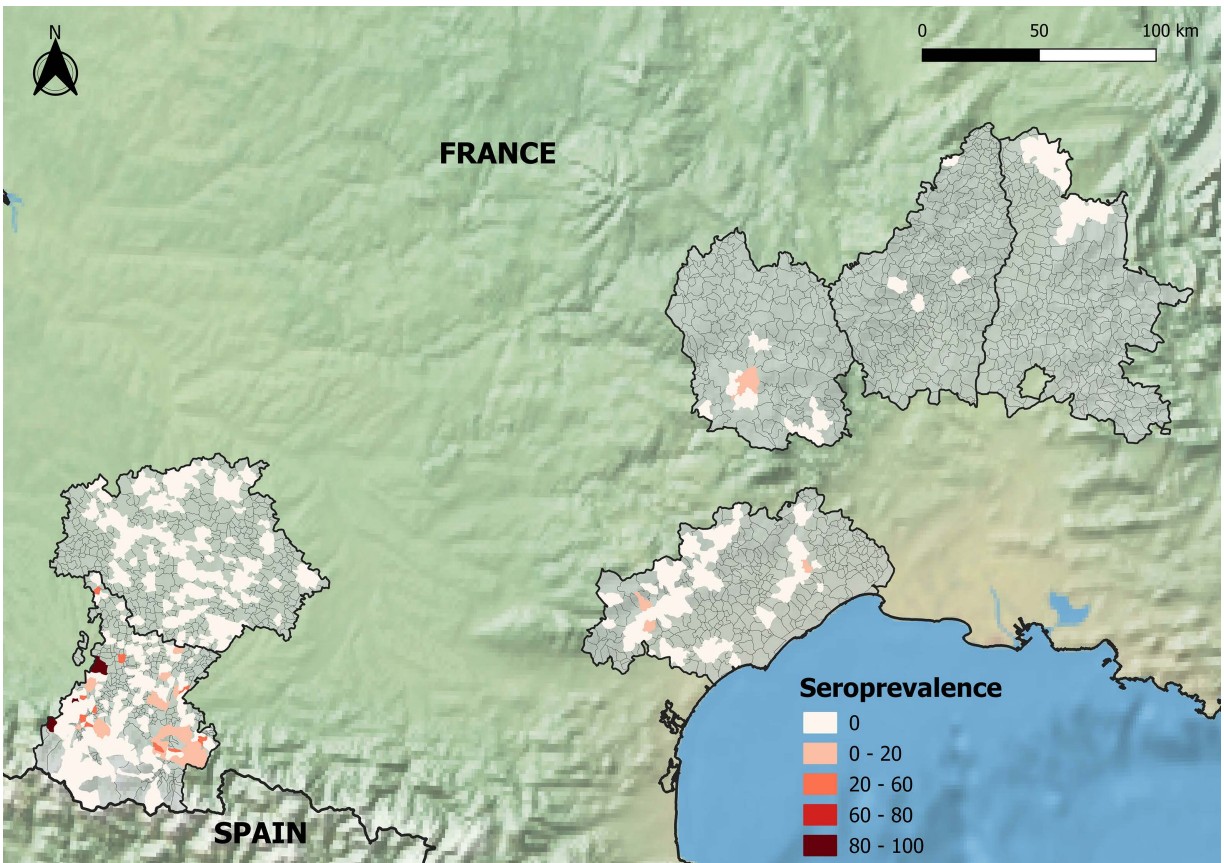

**Fig 3. Map of wildlife seroprevalences per municipality in the administrative departments that agreed to take part in the study (pale pink to dark red, colors indicate increasing seroprevalence; sampled but negative municipalities are shown in gray).** Background layers derived from Natural Earth (CC BY 4.0).

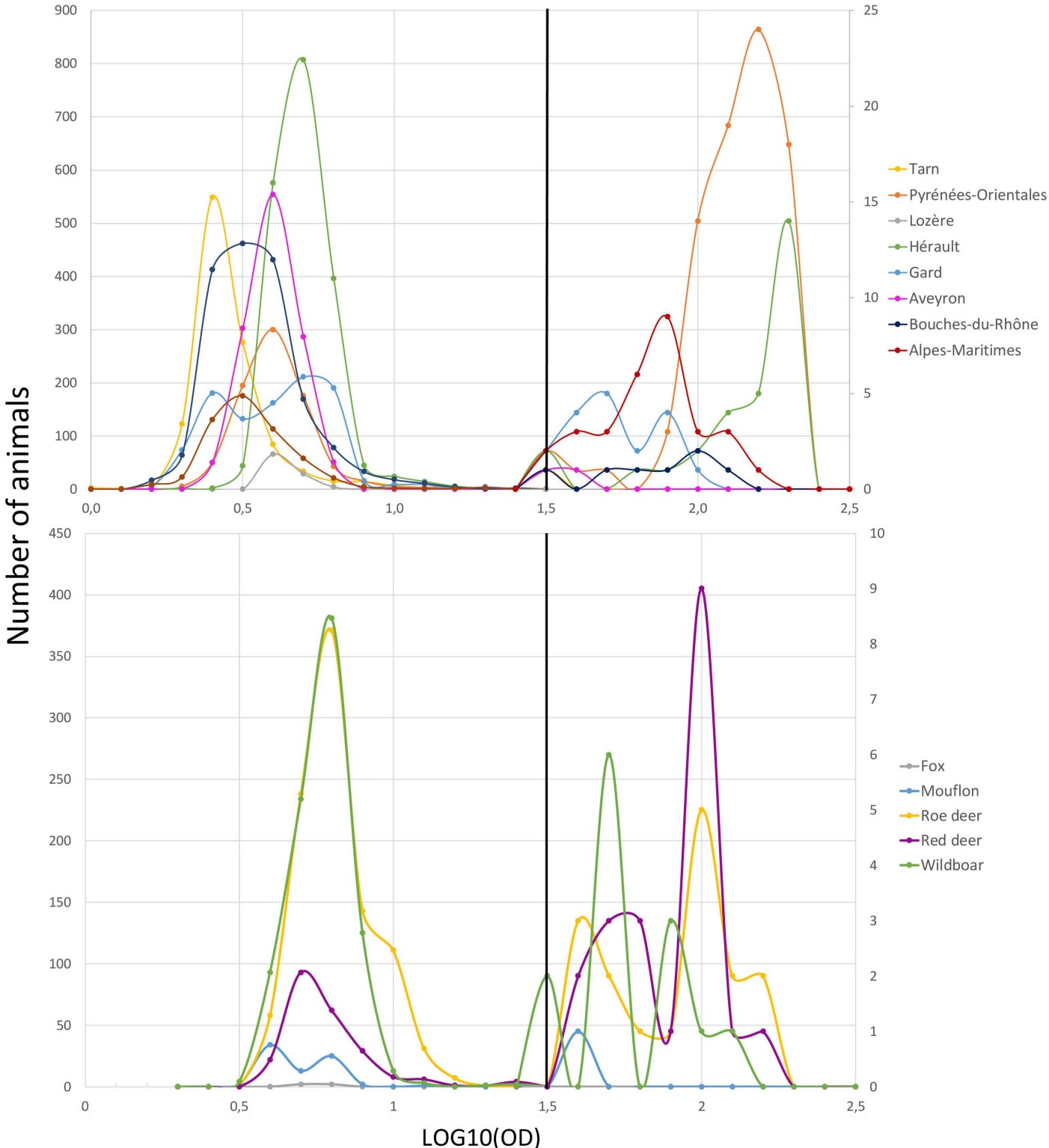

**Fig 4. Estimated density distributions of positive and negative samples across different administrative departments (A) and animal species (B), based on optical density measurements.** The curves on the left represent negative samples, those on the right to positive samples based on their optical density values. The curves illustrate the distribution profiles of optical density, highlighting differences across regions and species, and distinguishing positive from negative samples.

wildlife samples showed two distinct peaks, one peak aligning closely with the threshold and another with OD values (Fig 4B). This pattern remained similar in the different animal species. The OD value of the single positive mouflon found in Lozère was close to the threshold.

## Confirmation of positive results using pseudo-plaque reduction neutralization tests (PPRN)

A total of 59 ELISA-positive cattle sera (Alpes-Maritimes = 17, Bouches-du-Rhône = 3, Gard = 10, Hérault = 20, Pyrénées-Orientales = 9) and 40 ELISA-positive wildlife sera (Hautes-Pyrénées = 39, Lozère = 1), in addition to 5 positive and 10 negative controls (as presented above), were sent to a BSL4 laboratory for PPRN testing. The results of PPRN are given in S1 Table. Among controls, the 10 ELISA-negative samples were PPRN-negative, and the 4 positive controls were confirmed positive by PPRN up to 1/80 dilution and one at 1/20 dilution. In cattle, 39 out of the 59 ELISA-positive sera were confirmed with a minimum threshold of 1/20. Those that could not be confirmed mainly came from Alpes-Maritimes (n = 6/17), Bouches-du-Rhône (n = 3/3), Gard (n = 8/10) and a few from Hérault (n = 3/20). Concerning wildlife, 39 out of the 40 ELISA-positive sera were confirmed with a minimum threshold of 1/20, and most (n = 30/39) were still confirmed at 1/80 dilution.

There was a slight correlation between OD values provided by the ELISA test and the dilution thresholds (DT) provided by the PPRN test for cattle ($R^2 = 0.35$) and wildlife ($R^2 = 0.29$) (Fig 5A and 5B), suggesting that the higher the level of antibodies, the higher the dilution until which it is still possible to detect neutralizing antibodies. However, particularly in cattle, some samples with high OD values remained negative in PPRN or were only confirmed at low dilutions.

## Factors explaining seropositivity in animals

In cattle, age had a highly significant effect on animal seropositivity (P-value = 0.0004), indicating that the older the animal, the more likely it is to become infected and hence be seropositive. In addition, farms including a high proportion of natural open habitats had significantly more seropositive animals than locations where the other habitats predominated (P-value = 0.027). The variables "Sex" and " Closed Coniferous Forest " were also retained in the final model but

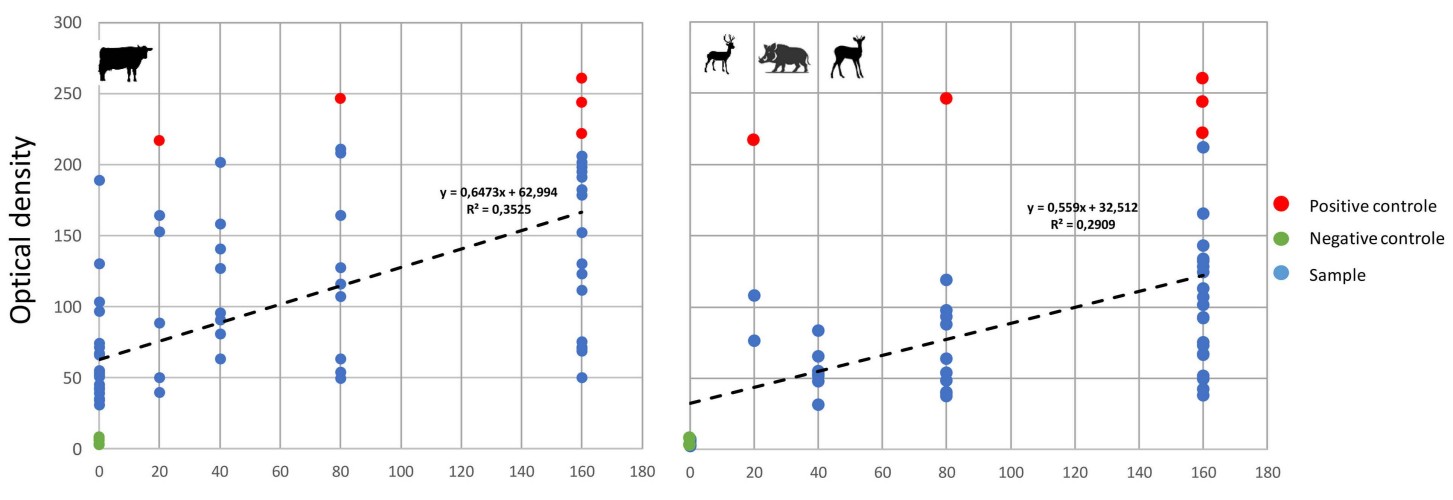

**Fig 5. Correlation between the optical density (OD) of sera samples in ELISA and the dilution threshold (DT) until which the neutralizing effect of antibodies in sera was still observed by PPRN, in cattle (A) and wildlife (B).** The red dots indicate reference positive controls (ELISA-positive Corsican sera), the green dots indicate negative ones (ELISA-negative sera from Corsica and the Netherlands).

their effect were not significant, fewer males were seropositive than females and locations showing a predominance of closed coniferous forests with a higher proportion of seropositive cattle (Table 3). Concerning random effects, substantial variance was observed at the level of farm in a given municipality and a given department (variance=47.473, std. dev.=6.890), and to a lesser extent at the level of the municipality in a department and at the level of the department, suggesting significant spatial heterogeneity in the serological status of farms even when they are located close to one another (Table 3). The variance explained by the breed of cattle was also low, suggesting that some breeds of cattle are unlikely to be infected through tick bites and consequently to be seropositive. The final model had an AIC of 775.2 and a BIC of 838.7, indicating an acceptable fit of the data to the model. The model's intercept was estimated to be −12.619 (p<2e-16), suggesting a very low baseline probability that an animal is seropositive.

In wildlife, fewer variables were retained in the final model and only Sex had a significant effect of the serological status of animals (P-value=0.0141), many more males being seropositive than females, the opposite of what was found in cattle. The proportion of "Open_Forest" in municipalities was retained but had no significant effect on the seropositivity of animals (P-value=0.6783. Similarly, the third synthetic climate variable "DIM 3", mainly explained by precipitation in the fall, was retained but had no significant effect on seropositivity (P-value=0.1749). Random effects captured inter-departmental variability, with an estimated variance of 3.914. These results highlight the contribution of local factors and individual characteristics to the probability of infection, thereby confirming the advantage of using mixed models in such contexts. The intercept of the model indicates both a baseline probability of a positive serological response and significant variations across geographical locations (departments). The AIC simulated for the model is 1 523.45, and the BIC is 1 540.78. These values indicate a trade-off between model fit and complexity, with lower values suggesting better model performance.

## Discussion

This work led to the first detection of antibodies against CCHFV in cattle and wildlife in mainland France, and identified areas where CCHFV circulates, at least in the French administrative departments in which the serological investigations were conducted. These novel results question the specificity of the ID Screen® CCHF DA ELISA we used in our study, i.e., the capacity of the test to only detect true positives. As mentioned above, by testing several sera samples from cattle, small ruminants and humans from CCHFV-free areas as well as sera from several different animal species to confirm the absence of nonspecific reactions, a specificity of 100% (CI95%: 99.8%−100%) was determined [29]. Nevertheless, one

**Table 3. Factors retained in the final multivariate logistic regression model for cattle and wildlife (depending on the model, empty rows mean the variables were not retained), including the indication for each retained variable, the tendency (estimated), and the significance of its effect (P-value).**

|  | Cattle |  |  | Wildlife |  |  |
|---|---|---|---|---|---|---|
|  | **Fixed effects** |  |  |  |  |  |
| *Predictors* | *Estimate* | *CI95%(+/-)* | *P value* | *Estimate* | *CI95%(+/-)* | *P value* |
| (Intercept) | −12.620 | 1.090 | **< 0.001** | −6.749 | 1.241 | **< 0.001** |
| Sex [M] | −0.224 | 0.390 | 0.57 | 0.982 | 0.400 | **< 0.05** |
| Age | 0.110 | 0.031 | **< 0.001** |  |  |  |
| Dim_3 |  |  |  | 0.127 | 0.093 | 0.17 |
| Closed coniferous forest | 0.036 | 0.021 | 0.09 |  |  |  |
| Open forest |  |  |  | 0.040 | 0.096 | 0.69 |
| Open natural habitat | 0.041 | 0.018 | **< 0.5** |  |  |  |
|  | **Random effect** |  |  |  |  |  |
|  | Farm:(Commune:DEP) |  |  | DEP |  |  |
| Variance | 47.473 |  |  | 3.914 |  |  |
| Std. Dev | 6.890 |  |  | 1.978 |  |  |

can speculate about potential cross-reactions with viruses genetically and antigenically close to CCHFV. A previous PPRN test carried out on ELISA-positive Corsican cattle sera showed a typical neutralizing effect of antibodies against CCHFV culture whereas no effect was observed with Hazara and Dugbe viruses, which belong respectively, to the same serogroup of CCHFV or a very close one (namely, the Nairobi sheep disease virus serogroup), [30]. Some of these Corsican sera were included in our study and produced identical ELISA and PPRN results to the other sera from the mainland that we tested; discrepancies in some ELISA-positive sera that were not confirmed by PPRN could be due to different target epitopes in the two tests (i.e., ELISA targets the nucleoprotein, while PPRN targets the glycoprotein) [33]. What is more, another study of cattle in Nigeria found no correlation between the results of the ID Screen® CCHF DA ELISA and the DUGV ELISA developed to detect specific antibodies against Dugbe virus, whereas cross-reactions were reported when immunofluorescence was used [34]. Another study in which one calf and one sheep were hyper immunized with inactivated Hazara virus demonstrated that the ID Screen® CCHF DA ELISA was the only CCHFV serological test able to detect these animals as negative ones and consequently to distinguish between antibodies against Hazara virus and CCHFV [35]. Similarly, another study showed that ruminants experimentally infected with Nairobi sheep disease virus were all negative for CCHFV antibodies when tested with the ID Screen® CCHF DAM ELISA [36]. Unless an unknown CCHF-like virus exists (sufficiently close to CCHFV to create cross-reaction in ELISA but sufficiently distinct from CCHFV to avoid detection by CCHFV but also panNairovirus PCRs), based on all these data, our serological results suggest CCHFV is currently circulating in mainland France. With the recent detection of the causative virus in local *Hyalomma* ticks [24], the presence of persistent antibodies in ruminants already confirmed the existence of an enzootic CCHFV transmission cycle between animals and ticks in France although human cases have not yet been reported.

In our study, CCHF epidemiological patterns appear to differ among the administrative departments sampled in southern mainland France. In Tarn, Aveyron, Lozère and Bouches-du-Rhône, seroprevalences were very low, with maximum one seropositive animal per farm. OD values of positive animals were close to the threshold, which can thus be interpreted either as false positive results, or as single infections resulting in seroconversion that might be old relative to the date the antibody was detected, with a possible decline of antibody titer, at least for the IgM that are also measured by the ELISA test used in this study. As the assumed CCHFV tick vector *H. marginatum* is predicted to be absent in these departments [23], in the present study, they were thus considered to be CCHFV-free areas. Single infections could be due to human-driven movement of domestic or wild ungulates from an endemic area where CCHFV already circulates, as already reported in other countries [26,37], or to the introduction of an infected *H. marginatum* tick vector at the immature stage through bird migrations or the introduction of lagomorphs for the purpose of hunting followed by the molting of this tick in an adult that might be able to parasitize and infect cattle locally [38–40]. Sporadic introductions of *H. marginatum* via trans-Mediterranean migratory birds used to be common in Europe and could explain most of single adult ticks found on horses in northern countries in summer, because increasing temperatures allow nymphs to molt in adult stages, but without the possibility for the tick to complete its entire development cycle and become established in these areas [32]. In Gard and Hérault, seroprevalences and OD values remained low, seropositive municipalities were sparse, and the number of seropositive animals per farm was still close to 1, suggesting a similar non-circulating epidemiological situation. However, in some farms, several animals were seropositive with high OD values, suggesting local CCHFV circulation with regular or recent infection of animals leading to persistent high virus titers. *H. marginatum* is known to be present in these departments [22,23], and probably explains such transmission. In Hérault, the fact of detecting a few seropositive wild boars, which are possible hosts for adult *H. marginatum*, also confirms the hypothesis of local foci of CCHFV circulation in domestic and wild ungulates. Finally, in Alpes-Maritimes and Pyrénées-Orientales, seroprevalences were significantly higher (7 and 9%, respectively) than in the other departments and spatial clusters were identified around some municipalities, with many farms hosting a large number of positive animals. Such seroprevalences are quite similar to those observed in cattle in regions with what are assumed to be comparable epidemiological systems, for instance, the French island of Corsica (13.3%, 95%CI: 10.2%−17.3%) [30], southern Italy (1.89%, 95%CI: 1.12–3.1) [21] or Central Macedonia

in Greece (7%, 95%CI: 5%−10%). However, these seroprevalences remain very low compared to those observed in cattle in some CCHF-endemic African countries such as Mauritania (69%) or Mali (66%), where the epidemiological transmission cycles are different and involve different tick vectors, with high parasitic loads on ruminants and different host preferences [41,42]. Seroprevalences detected in wildlife in Hautes-Pyrénées (5.03% in wild boars and 8.16% in cervids) were slightly higher than those observed in the same species on the other side of the Spanish border, in Catalonia [19], but they remain lower than those detected in central Spain where human CCHF cases have been regularly reported since 2016. However, in central Spain, the epidemiological transmission cycle likely involved the tick *H. lusitanicum* rather than (or in addition to) *H. marginatum* and the results are consequently not fully comparable. In our study, OD values were medium to high in cattle in Alpes-Maritimes and Pyrénées-Orientales, as well as in wildlife in Hautes-Pyrénées, with similar patterns to those observed in Corsican ruminants [30]. All these results agree with frequent and active CCHFV transmission events between animal hosts and tick vectors in such departments of mainland France, although further field monitoring studies are needed to identify the precise spatiotemporal dynamics of virus transmission. The seroprevalence in roe deer in this study, as high as that in red deer, is the highest ever reported in the literature even in South-Central Spain where roe deer were found to be infested with infected *Hyalomma lusitanicum* [43]. It is also worth mentioning that, despite the fact that CCHFV antibodies have been reported in other populations of mountain ungulates [19], this is the first time a seropositive mouflon (*Ovis musimon*) has been detected in France, namely in Lozère. As it was a single individual, it remains uncertain if this was a false positive animal or just a single infection in the population following introduction of a *Hyalomma* tick (see above).

Although *H. marginatum* is very abundant and widely distributed in Pyrénées-Orientales, in line with active CCHFV transmission in this department, this species of tick was not found in Alpes-Maritimes in two tick collection campaigns conducted on horses in 2017 [44]. A similar discrepancy was found in Hautes-Pyrénées where seroprevalence in wildlife is high, whereas according to the distribution model developed by Bah et al. the tick *H. marginatum* is predicted to be absent in this department [23]. Several hypotheses could explain these situations. First, as the invasion of new French areas by *H. marginatum* in France is still underway under climate change, one may assume that these zones have now become suitable for its establishment, provided it can spread from an endemic area and create populations that are abundant enough to be detected by sampling. Another hypothesis is that an additional tick vector to *H. marginatum* exists in these departments. CCHF is a complex disease and, among its worldwide distribution, its causative agent can be transmitted by different tick species belonging to the *Hyalomma* genus but also to other genera like *Rhipicephalus, Amblyomma,* and possibly *Dermacentor* [10,26]. Recently reported as a likely vector of CCHFV in Spain [43], *H. lusitanicum* could be a potential candidate for France. Historical reports from France mentioned it in the western Atlantic region and Pyrénées-Orientales, which both surround Hautes-Pyrénées, and in the eastern part of France, in Bouches-du-Rhône and Var near Alpes-Maritimes [45,46]. It thus appears to be much more selective in its choice of vertebrate animal hosts than *H. marginatum*, with a likely preference for the immature stages of lagomorphs, especially wild rabbits [47,48]. As rabbit populations in the South of France have declined significantly since the 1970s due to the epizooty of myxomatosis and more recently to rabbit hemorrhagic disease [39,40], we hypothesized that this species had disappeared from the territory [26]. However, a small population of *H. lusitanicum* was recently detected in wild rabbit burrows in Bouches-du-Rhône (Stachurski, pers. comm.). This could be a residual population or a new population resulting from rabbit introduced from Spain [40]. The extent of the distribution of this species of tick in France is not known and further investigations are thus required in other French departments. If in the future, the establishment of *H. lusitanicum* in Hautes-Pyrénées is confirmed, this might explain the high CCHFV seroprevalence measured in roe deer and red deer, which are likely hosts for the adult stages of *H. lusitanicum* [46,47]. In Alpes-Maritimes, roe deer also frequent in areas identified as CCHFV circulation hotspots, but further field investigations are needed of tick distribution along with a serological survey of wild animals to test these hypotheses. Another tick species that has frequently been collected on horses in Alpes-Maritimes was *Rhipicephalus bursa*, which has been proposed as a candidate vector for CCHFV although its vector competence

has never been demonstrated [15]. This tick species is also frequently encountered in sympatry with *H. marginatum* on horses and cattle in other southern French departments such as Pyrénées-Orientales, with similar seasonal activity at least during its adult stages. In regions where local transmission of CCHFV has been demonstrated, it would be interesting to investigate this tick species to assess its potential role in CCHFV transmission.

As *H. marginatum* is considered to be the only tick vector of CCHFV in France [24], the presence of a CCHFV tick vector, namely *H. marginatum* [26] was tested in our study to explain CCHFV seropositivity in cattle and wildlife, under the assumption that the presence of a tick vector is one of the main environmental factors driving natural enzootic transmission of CCHFV although tick vectors are not necessarily infected everywhere. Using the presence of *H. marginatum* predicted by Bah *et al.* [23], our model failed to detect any effect of the presence of this tick. However, seropositivity in cattle was higher in areas with a high proportion of natural open habitats (i.e., shrublands in Mediterranean regions) and coniferous forests, and for wildlife, Open Forest was retained in the final model even though the effect of this variable was not significant. Natural open habitats have already been identified as the most suitable conditions for the presence of *H. marginatum* [23], and likely also identify optimal conditions for cattle being infested by CCHFV tick vectors. However, other parameters than tick vector distribution may influence the efficiency of CCHFV transmission [26], such as the presence and the abundance of CCHFV-amplifying hosts, as well as simultaneous encounters between these hosts and the infected ticks. The fact that coniferous and open forests were signaled by both our models could reflect the important role played by wildlife, particularly cervids, in the CCHFV enzootic transmission cycle, as these animal species mainly colonize forested areas and are suspected to replicate CCHFV and reinfect new tick vectors in Spain [43]. In addition, under the hypothesis of another tick vector like *H. lusitanicum*, and apart from the fact that abundant populations of cervids could amplify its populations, such forested areas may also provide suitable conditions for the survival and development of this tick species, as proposed by Valcárcel et al. [47]. In this case, cattle in farms surrounded by such habitats would be likely to encounter infected tick vectors. In these departments, further research is therefore needed on the ecological niche of *H. marginatum* and other candidate tick vectors, including also local interactions and likely associations between animals and tick species.

Apart from environmental conditions, anthropogenic factors related to farming practices were also tested to explain cattle seropositivity. Contrary to our assumption that the type of cattle farming (dairy, suckling, mixed, or cattle raised for recreational purposes) influence the exposure of animals to tick vectors, in our study, this variable had no effect on cattle seropositivity. In the South of France, most cattle are raised for meat production, although in areas such as the Camargue some of them are also raised for recreational purposes, especially local Camargue bullfight and associated activities [48]. We consequently had an over-representation of suckling cattle compared to the other categories. In addition, it was sometimes complicated to attribute a breed to a category, especially for mixed uses. Finally, within the prophylaxis process defined in France, the blood of dairy cattle is not sampled, milk is tested for disease control. As a consequence, this category was even more under-represented compared to suckling cattle in our study. Thus, based on the current study design, we cannot conclude on such an effect. Another anthropogenic factor that was tested in our model concerned contacts between cattle and other animal species, as other species may have different abilities for replicating CCHFV, for example by contaminating tick vectors, and increasing CCHFV transmission in the vicinity of cattle [49,50]. No significant effect on cattle seropositivity was found for this variable, but neither were we able to test contacts with wildlife as this information was not available in the databases we used in the present study. This possible effect? should be investigated in future field surveys by questioning farmers directly about their fencing practices as well as about their observations of wildlife around their farms. Questionnaires distributed to farmers could also collect information concerning other practices such as their use of anti-parasitic products (insecticides, acaracides, or anthelmintics with ivermectine that is also efficient against ectoparasites such as ticks), a parameter we were also unable to include in our study due to lack of data. All these treatments are intended to reduce tick infestation and thus decrease the exposure of cattle to infected tick vectors. This phenomenon was reported in cattle in Pakistan [50] and was

considered to be a reliable preventive strategy to avoid CCHFV transmission when animals were moved from Sudan to Saudi Arabia [37]. Although no anthropogenic factors tested in our model were shown to have an effect on cattle seropositivity, the very fact that a large proportion of data variance was explained by the farm as a random effect, confirmed heterogeneity among farms, even farms located in the same municipality, which could be explained either by the farm environment or by farming practices. A last practice that is widespread in the South of France, especially for suckling cattle, and needs to be further investigated is the temporary transfer of cattle (transhumance) to spring and summer pastures during the period the adult stages of *Hyalomma* ticks are active. As these pastures are usually located far from the farmstead and are consequently not included in the description of the environment surrounding farms, this may represent a limitation of our current cattle model, i.e., its ability to identify habitats where cattle are most likely to be infested by CCHFV tick vectors. Studies conducted in Africa and Asia reported higher CCHFV seroprevalence in animals grazing in pastures than in animals fed from troughs [49,50], and even higher prevalence in nomadic herds that cover long distances [51].

Finally, individual factors were tested to explain animal seropositivity in our survey. The sex of animals was retained in final models for both cattle and wildlife; females presented higher seroprevalence than males in cattle, while the reverse was the case in wildlife. This observation is quite rare, only two studies of cattle in Malawi and South Africa reported higher seroprevalence in females than in males [28,50]. In our study, the over-representation of suckling cattle (see above) may have resulted in a bias favoring female over males, as reproductive females are usually kept longer than males that are sold for fattening or directly slaughtered when a few months old. Considering this bias, we cannot conclude on a reliable effect of sex on the probability of cattle being seropositive. In the case of wildlife, the fact the male's home range is larger than that of females [52], particularly during the reproductive period, may multiply the opportunities for them to visit areas infested by CCHFV tick vectors. Another individual factor that significantly affected cattle seropositivity in our study was the age of the animals. As already documented in many serological surveys on CCHF in domestic ungulates, older animals have a higher probability of being seropositive as a result of their longer exposure to infected tick vectors in successive years and the persistence of CCHFV antibodies for several years in most ungulates [53]. Conversely, age had no effect on wildlife seropositivity. However, in our study, age was categorized in only three classes and assignment to an age category depends on the subjectivity of hunters, who may have difficulty differentiating adults and subadults. In addition, cattle breed as well as species of animal wildlife were also tested as random and fixed effects, respectively, on seropositivity. None of these variables had a significant effect and were consequently retained in our best models. This result is interesting because some studies conducted in Africa confirmed the apparent resistance of certain breeds of cattle, particularly native breeds, against tick infestation, and hence CCHFV transmission [54], but this tendency was not systematically highlighted [27], and in some cases was just the opposite [49]. In France, given that the assumed tick vector *H. marginatum* has only recently colonized the territory, it appears that no local breed has had time to develop adaptive resistance to tick infestation, in agreement with our results. Concerning wildlife, the absence of an effect of the species of animal suggests that wild boar, roe deer and red deer were equally infested by CCHFV tick vectors, at least in Hautes-Pyrénées, where most of seropositive animals were detected. This underlines the importance of further surveys of tick vectors' distribution and trophic preferences, as well as large-scale serological investigations of the different species of wildlife in the other departments.

## Supporting information

**S1 Fig. Directed Acyclic Graph (DAG) representing the hypothesised causal relationships between explanatory variables (individual, anthropogenic and environmental) and CCHFV seropositivity in animals.** This conceptual framework was used to guide the selection of variables in multivariable hierarchical models and to reduce the risk of overadjustment or inclusion of collinear variables.
(TIF)

**S1 Table. Comparison of ELISA and PPRNT results for all samples.**
(PDF)

## Acknowledgments

The authors thank the departmental veterinary laboratories and departmental hunting federations for their help in providing samples. We would also like to thank the management of health protection groups in the administrative departments for their assistance in obtaining breeders' agreements. We are grateful to all the cattle owners who collaborated with us on this study. We would like to thank the French Ministry of Agriculture for giving us access to the national database for livestock identification. Finally, we thank CIRAD support services for assistance in the process of protecting cattle owners' personal data.

## Author contributions

**Conceptualization:** Célia Bernard, Vladimir Grosbois, Benoit Combes, Matthieu Bastien, Laure Guerrini, Philippe Holzmuller, Laurence Vial.

**Formal analysis:** Célia Bernard, Andrea Apolloni, Laurence Vial.

**Investigation:** Célia Bernard, Armelle Peyraud, Phonsiri Saengram, Olivier Ferraris, Valentin Chauvin.

**Resources:** Ferran Jori, Eva Faure, Nicolas Keck, Raphaëlle Pin, Loic Comtet.

**Writing – original draft:** Célia Bernard, Laurence Vial.

**Writing – review & editing:** Célia Bernard.

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
