## [Decision Letter · Decision Letter 0]

4 Jun 2025

Dear Dr. Bernard,

Thank you for submitting your manuscript to PLOS ONE. After careful consideration, we feel that it has merit but does not fully meet PLOS ONE’s publication criteria as it currently stands. Therefore, we invite you to submit a revised version of the manuscript that addresses the points raised during the review process.

We look forward to receiving your revised manuscript.

Kind regards,

Bekir Oguz

Academic Editor

PLOS ONE

2. Please update your submission to use the PLOS LaTeX template. The template and more information on our requirements for LaTeX submissions can be found at http://journals.plos.org/plosone/s/latex"

 [The authors thank the funders who made this work possible: French Ministry of Agriculture—General Directorate for Food (DGAl, grant agreement: SPA17 number 0079-E), European Funds for Regional Development (FEDER, Grand-Est), French Establishment for Fighting Zoonoses (ELIZ) and the Association Nationale Recherche Technologie (ANRT, grant agreement number: 2019-1145)]. 

4.  We note that Figures 1-3 in your submission contain [map/satellite] images which may be copyrighted. All PLOS content is published under the Creative Commons Attribution License (CC BY 4.0), which means that the manuscript, images, and Supporting Information files will be freely available online, and any third party is permitted to access, download, copy, distribute, and use these materials in any way, even commercially, with proper attribution. For these reasons, we cannot publish previously copyrighted maps or satellite images created using proprietary data, such as Google software (Google Maps, Street View, and Earth). For more information, see our copyright guidelines: http://journals.plos.org/plosone/s/licenses-and-copyright.

1. You may seek permission from the original copyright holder of Figures 1-3 to publish the content specifically under the CC BY 4.0 license.   

Additional Editor Comments (if provided):

Reviewers' comments:

Reviewer's Responses to Questions

**Comments to the Author**

1. Is the manuscript technically sound, and do the data support the conclusions?

Reviewer #1: Yes

Reviewer #2: Yes

2. Has the statistical analysis been performed appropriately and rigorously?

Reviewer #1: Yes

Reviewer #2: Yes

3. Have the authors made all data underlying the findings in their manuscript fully available?

Reviewer #1: Yes

Reviewer #2: No

4. Is the manuscript presented in an intelligible fashion and written in standard English?

Reviewer #1: Yes

Reviewer #2: No

Reviewer #1: Crimean-Congo Hemorrhagic Fever Virus (CCHFV) infections are widely distributed across Africa, Eastern Europe, the Middle East, and Asia, largely mirroring the natural habitats of its principal vector, Hyalomma spp. ticks. Rural populations, particularly those engaged in livestock farming, are at increased risk of infection through tick bites or direct contact with viremic animals. In recent years, CCHFV has also emerged as a growing public health concern in Europe. The detection of the virus in ticks, seropositivity in vertebrate hosts, and reported clinical cases in more than ten European countries underscore the virus’s expanding geographic footprint. Notably, new emergence and local transmission have been documented in countries such as Bulgaria, Greece, Spain, and Portugal, highlighting the urgent need for region-specific surveillance and control strategies.Recent studies conducted in Corsica and Spain further demonstrate the expanding range of CCHFV. In France, serological findings suggest that the virus may be present in various tick species, domestic animals, and potentially in humans. These findings imply that the virus, once believed to be confined to specific geographic regions, could continue spreading northward within Europe. In this context, the current study contributes valuable insights by emphasizing the importance of updated epidemiological data. Wildlife surveillance, when utilized as an early warning system, can help trace the virus’s circulation before clinical cases appear. In the near future, comprehensive studies integrating both serological and virological methods will be essential to accurately map the true distribution and transmission dynamics of CCHFV across the continent.

Reviewer #2: The paper, First detection of Crimean Congo Hemorrhagic Fever antibodies in cattle and wildlife of southern continental France: investigation of explicative factors documents the detection of CCHF in the French mediterranean region in 2008 to 2022 using serological tests. It also reports animal-level and ecological factors that are associated with exposure in livestock and wildlife.

The main comments I have are:

1. Statistical modelling – given that many variables were offered for statistical modelling using hierarchical multivariable regression model, a formal approach for identifying orthogonal variables for inclusion must be used to improve the efficiency of the analysis. This will include using causal web models, directed acyclic curves etc. I recommend that this approach is included in the analysis.

2. The language should be reviewed to improve its presentation in consistent English language

Other comments:

Line 43: CCHF has significant public health impacts in regions where the disease is endemic. So even if it does not expand its geographical range, the disease still has high public health significance. I suggest rewording this statement to reflect that fact

Line 69 and other places, I suggest replacing my study with our study

Line 89 and 104: say humans are infected not contaminated

Line 91: Is WHO a reference?

Line 143/148: delete detecting in the title since it is already implied in the sentence

Line 176: Start the line with text (45). Same as line 209

Line 529: use spatial coverage

Line 530. Put a full stop after level

Line 587: Use a different work instead of demonizing

Line 651: this should be multivariable model not multivariate model

**Do you want your identity to be public for this peer review?** For information about this choice, including consent withdrawal, please see our Privacy Policy

Reviewer #1: No

Reviewer #2: No

---

## [Author Response · Author response to Decision Letter 1]

30 Jul 2025

Response to Reviewers – Manuscript PONE-D-25-20397

We thank the Academic Editor and the reviewers for their constructive and insightful comments. We have carefully revised the manuscript to address all points raised. Below, we provide detailed responses to each comment. All changes have been made in the revised manuscript and highlighted using tracked changes.

Reviewer #1

We thank the reviewer for their thoughtful overview and contextual framing of the current epidemiological situation of Crimean-Congo Hemorrhagic Fever Virus (CCHFV) in Europe. We fully agree that future integrated studies combining serological and virological approaches will be key to better mapping and understanding the dynamics of CCHFV transmission in Europe.

All the changes suggested in the document have been taken into account.

Comment : As stated in the discussion section, lines : 367 -368''other tick species may also be effective.'' Therefore, in the introduction section, brief information can be given about tick diversity in regions where the disease has been detected, especially in Spain, and about possible vectors other than hyalomma spp. in France. It will be more informative for the readers.

Response: We would like to thank the reviewer for his comment. We decided to add a sentence to be more informative. “Other tick species such as Rhipicephalus bursa and Dermacentor marginatus have occasionally been found carrying CCHFV in southwestern Europe, but their role in transmission remains uncertain. In France, despite the presence of multiple tick genera, only H. marginatum has been found positive to date”.

Reviewer #2

We thank the reviewer for their constructive feedback and suggestions to improve the manuscript. Below, we provide point-by-point responses to each comment.

Comment: 1. Statistical modelling – given that many variables were offered for statistical modelling using hierarchical multivariable regression model, a formal approach for identifying orthogonal variables for inclusion must be used to improve the efficiency of the analysis. This will include using causal web models, directed acyclic curves etc.

Response: Thank you for this suggestion. In response, we constructed a directed acyclic graph (DAG) to represent the hypothesized causal relationships between the exposure, explanatory variables, and CCHFV seropositivity in both livestock and wildlife. This DAG, based on literature and expert knowledge, was used to identify potential confounding structures and guide the variable selection process in the hierarchical multivariable models. The DAG is now provided as Supplementary Figure S1. In parallel, we performed correlation tests on the variables retained in our final models (after the dredge step and selection of the best-fitting model). These tests showed that the variables were not significantly correlated. In addition, mean comparison tests indicated that the group means were significantly different.

Comment: 2. The language should be reviewed to improve its presentation in consistent English.

Response:

The entire manuscript has been thoroughly reviewed by a native English speaker and edited to ensure consistency, clarity, and correct grammar throughout.

Comment: Line 43: CCHF has significant public health impacts in regions where the disease is endemic. So even if it does not expand its geographical range, the disease still has high public health significance. I suggest rewording this statement to reflect that fact.

Response:

We agree and have reworded this sentence to reflect that CCHF remains a significant public health concern regardless of its geographic expansion.

Comment: Line 69 and other places: replace "my study" with "our study"

Response:

All instances of “my study” have been replaced with “our study” for consistency and appropriate authorship tone.

Comment: Line 89 and 104: say "humans are infected" not "contaminated"

Response:

We have replaced “contaminated” with “infected” in both instances (lines 89 and 104), which is the accurate term for human exposure to CCHFV.

Comment: Line 91: Is WHO a reference?

Response:

We have changed the reference in the manuscript.

Comment: Line 143/148: delete "detecting" in the title since it is already implied in the sentence

Response:

The term “detecting” has been removed from the respective sentences for conciseness and clarity (lines 143 and 148).

Comment: Line 176 and 209: Start the line with text instead of a number (e.g., "(45)")

Response:

We have revised the formatting to ensure that each sentence starts with text before the reference, as suggested.

Comment: Line 529: use "spatial coverage"

Response:

We have replaced the original term with “spatial coverage” in line 529.

Comment: Line 530: put a full stop after "level"

Response:

A full stop has been added after “level” (line 530).

Comment: Line 587: use a different word instead of "demonizing"

Response:

We have replaced “demonizing” with a more appropriate term (“stigmatizing”) to improve tone and clarity (line 587).

Comment: Line 651: use "multivariable model" instead of "multivariate"

Response:

We have corrected the term to “multivariable model” in line 651 to accurately reflect the modelling approach.

Journal Requirements

1. Formatting

Response: The manuscript has been updated to match the PLOS ONE style and formatting requirements. File names have also been adjusted accordingly.

2. LaTeX template

Response: We acknowledge the request to format our manuscript using the PLOS LaTeX template. However, due to technical constraints, we were unfortunately not able to implement this formatting for the current version of the revised submission. We hope the current formatting remains acceptable for evaluation purposes, and we remain available to reformat the manuscript using the template if required at a later stage.

3. Financial disclosure update

Response: We have added the required sentence:

This has been added to both the cover letter and the manuscript.

4. Figures 1–3 – Copyright compliance

Response: The maps used in Figures 1–3 were generated from publicly available data sources (e.g., Natural Earth, BD Forêt, CORINE Land Cover). We confirm that no copyrighted images (e.g., Google Maps) were used. All sources have been cited in the figure legends, and figures comply with the CC BY 4.0 license. Alternatively (if needed), reworked figures are provided to ensure compliance.

5. Reference check

Response: We have reviewed all references and confirm that none are retracted. The reference list has been updated to ensure completeness and accuracy.

---

## [Editor Report · Decision Letter 1]

4 Aug 2025

Dear Dr.  Bernard,

We look forward to receiving your revised manuscript.

Kind regards,

Bekir Oguz

Academic Editor

PLOS ONE

Journal Requirements:

Additional Editor Comments:

Dear author, your article requires revision. Please make the required corrections and re-upload. Best regards.

---

## [Author Response · Author response to Decision Letter 2]

19 Aug 2025

We have carefully revised the manuscript according to the journal’s requirements. Specifically:

We reviewed the reference list to ensure completeness and accuracy. No retracted references were identified.

We revised the manuscript according to the PLOS ONE Word template.

We uploaded the revised figures to the PACE tool to ensure they meet formatting standards.

---

## [Editor Report · Decision Letter 2]

22 Aug 2025

First detection of Crimean Congo Hemorrhagic Fever antibodies in cattle and wildlife of southern continental France: investigation of explanatory factors

PONE-D-25-20397R2

Dear Dr. Bernard,

We’re pleased to inform you that your manuscript has been judged scientifically suitable for publication and will be formally accepted for publication once it meets all outstanding technical requirements.

Kind regards,

Bekir Oguz

Academic Editor

PLOS ONE

Additional Editor Comments (optional):

Dear Author,

The article has been edited in accordance with the feedback provided by the reviewers. I would say that it is acceptable as it is.

Best regards
---

## [Editor Report · Acceptance letter]

PONE-D-25-20397R2

PLOS ONE

Dear Dr. Bernard,

I'm pleased to inform you that your manuscript has been deemed suitable for publication in PLOS ONE. Congratulations! Your manuscript is now being handed over to our production team.

Kind regards,

on behalf of

Professor Bekir Oguz

Academic Editor

PLOS ONE